# An Atom-Centric Perspective on Stubborn Sets

**Gabriele Röger, Malte Helmert, Jendrik Seipp, Silvan Sievers**
University of Basel
Basel, Switzerland
{gabriele.roeger,malte.helmert,jendrik.seipp,silvan.sievers}@unibas.ch

## Abstract

Stubborn sets are an optimality-preserving pruning technique for factored state-space search. Their applicability in classical planning is limited by their computational overhead. We describe a new algorithm for computing stubborn sets that is based on the state variables of the state space, while previous algorithms are based on its actions. Typical factored state spaces tend to have far fewer state variables than actions, and therefore our new algorithm is much more efficient than the previous state of the art, making stubborn sets a viable technique in many cases where they previously were not.

*An archival version of this paper has been published at SoCS 2020 (Röger et al. 2020a).*

## Introduction

Heuristic search is a common approach for classical planning. Especially in *optimal* planning, the search suffers from a state explosion problem that occurs if states can be reached by applying the same actions in different orders. Even with close-to-perfect heuristics, the number of nodes that must be explored by pure heuristic search (only relying on node expansions and an admissible heuristic) can grow exponentially in the size of the task (Helmert and Röger 2008). Hence, search algorithms are often enhanced with pruning techniques that reduce the size of the explored state space.

One family of such pruning techniques is *partial order reduction*, which allows the search to ignore some paths to the goal by not considering all permutations of the actions. Intuitively, the idea is to avoid interleaving the solution of independent subproblems but instead solving one subproblem after the other. Partial order reduction was originally introduced by Valmari (1989) for Petri nets in the context of computer-aided verification. Alkhazraji et al. (2012) transferred his concept of *strong stubborn sets* to classical planning. Later on, Wehrle and Helmert (2014) generalized them with more fine-grained criteria that are still sufficient for optimality-preserving pruning. With suitable decisions at certain choice points, strong stubborn sets strictly dominate the *expansion core method* (Chen and Yao 2009; Wehrle and Helmert 2012), a partial order reduction technique introduced earlier for planning (Wehrle et al. 2013).

A stubborn set for a state is a set of actions such that all other actions can safely be ignored at its expansion. The concept is inherently action-centric and so are the underlying definitions and algorithms. In this paper, we adopt a more atom-centric perspective on their computation, which gives rise to a significantly faster algorithm. As an additional enhancement, we also contribute a new atom selection strategy, which has a tendency to produce smaller stubborn sets and leads to more pruning in our experiments.

## Background

We consider SAS$^+$ planning tasks (Bäckström and Nebel 1995), extended with non-negative action costs. A task is defined over a finite set $\mathcal{V}$ of *variables*, each associated with a finite domain $\mathcal{D}(v)$. A pair $(v, d)$ with $v \in \mathcal{V}$ and $d \in \mathcal{D}(v)$ is called an *atomic proposition*, or *atom* for short, and we use $\mathcal{P}$ to denote the set of all atomic propositions (over an implicit set of variables $\mathcal{V}$). We call all atoms $(v, d')$ with $d' \in \mathcal{D}_v \setminus \{d\}$ the *siblings* of atom $(v, d)$.

A *partial state* $s$ maps every variable $v$ from a set $vars(s) \subseteq \mathcal{V}$ to a value $s[v]$ from $\mathcal{D}(v)$. If $vars(s) = \mathcal{V}$, we call $s$ a *state*. When it is suitable, we also consider a partial state $s$ as the set of atoms $\{(v, s[v]) \mid v \in vars(s)\}$ and write $(v, d) \in s$ for $s[v] = d$.

A *task* is given as a tuple $\Pi = \langle \mathcal{V}, \mathcal{A}, s_\mathrm{I}, s_\mathrm{G} \rangle$ where $\mathcal{V}$ is the finite set of variables, $\mathcal{A}$ a finite set of *actions*, $s_\mathrm{I}$ a state called the *initial state* and $s_\mathrm{G}$ a partial state called the *goal*. Each *action* $a \in \mathcal{A}$ is defined by its *cost* $c(a) \in \mathbb{R}_0^+$ and two partial states $pre(a)$ and $eff(a)$ called its *precondition* and *effect*. If $(v, d) \in eff(a)$ for some atom $(v, d)$, we say that action $a$ *achieves* $(v, d)$. If $(v, d) \in pre(a)$, we say $a$ *depends on* $(v, d)$. W.l.o.g. we require that no action both depends on and achieves the same atom.

An action $a$ is *applicable* in state $s$ if $pre(a) \subseteq s$. Then the *successor state* $s'$ is given as $s'[v] = eff(a)[v]$ for all $v \in vars(eff(a))$ and $s'[v] = s[v]$ for all other variables. Slightly abusing notation, we write $a(s)$ for the successor state resulting from applying action $a$ in state $s$.

A *goal state* is a state $s$ with $s_\mathrm{G} \subseteq s$. A *plan* is a sequence of actions that are subsequently applicable in $s_\mathrm{I}$ and where the resulting state is a goal state. The cost of a plan is the sum of the individual action costs. A plan is *optimal* if it has minimal cost among all plans. Wehrle and Helmert (2014) pointed out that for correct pruning it is sometimes impor-

tant to only consider so-called *strongly optimal* plans, which are optimal plans with a minimal number of 0-cost actions among all optimal plans. If there is no plan for a task, the task is *unsolvable*. The aim of *optimal planning* is to find an optimal plan or to prove that the task is unsolvable.

Strong stubborn sets aim to prune permuted plans from the search. On a lower level, the permutation of actions is related to the following notion of *interference*.

**Definition 1** (interference, Wehrle and Helmert 2014). *Let $a_1$ and $a_2$ be actions and let $s$ be a state of a planning task $\Pi$. We say that $a_1$ and $a_2$ interfere in $s$ if they are both applicable in $s$, and*

- *$a_1$ disables $a_2$, i.e., $a_2$ is not applicable in $a_1(s)$, or*
- *$a_2$ disables $a_1$, or*
- *$a_1$ and $a_2$ conflict in $s$, i.e., $a_2(a_1(s))$ and $a_1(a_2(s))$ are both defined but differ.*

If two actions that are both applicable in a state $s$ do *not* interfere in $s$, we can apply them in any order and will in both cases reach the same state.

The second relevant notion are *necessary enabling sets*. These are related to disjunctive action landmarks (Helmert and Domshlak 2009), which are sets of actions of which at least one must be applied in every plan. Similarly, necessary enabling sets are sets of actions of which at least one must be applied *before a given action is applied* in every action sequence from a *given set*.

**Definition 2** (necessary enabling set, Wehrle and Helmert 2014). *Let $\Pi$ be a planning task, let $a$ be one of its actions, and let Seq be a set of action sequences applicable in the initial state of $\Pi$.*

*A necessary enabling set for $a$ and Seq is a set $N$ of actions such that every action sequence in Seq which includes $a$ as one of its actions also includes some action $a' \in N$ before the first occurrence of $a$.*

For this paper, we build on the generalized definition of strong stubborn sets by Wehrle and Helmert (2014) but for clarity we omit the concept of *envelopes*, which permit to safely ignore some actions. Empirically, the known methods for exploiting envelopes did not provide much benefit (Wehrle and Helmert 2014), and they can easily be re-integrated in our work.

**Definition 3** (strong stubborn set). *Let $s$ be a state of planning task $\Pi = \langle \mathcal{V}, \mathcal{A}, s_I, s_G \rangle$ and let $\Pi_s = \langle \mathcal{V}, \mathcal{A}, s, s_G \rangle$. A strong stubborn set in $s$ is a set $A \subseteq \mathcal{A}$ of actions that satisfies the following conditions.*

*If $\Pi_s$ is unsolvable or $s$ is a goal state, then every $A$ is a strong stubborn set. Otherwise, let Opt be the set of strongly optimal plans for $\Pi_s$ and let $\mathcal{S}_{Opt}$ be the set of states that are visited by at least one plan in Opt. The following conditions must be true for $A$ to be a strong stubborn set.*

- C1 *$A$ contains at least one action from at least one plan from Opt.*
- C2 *For every $a \in A$ that is not applicable in $s$, $A$ contains a necessary enabling set for $a$ and Opt.*
- C3 *For every $a \in A$ applicable in $s$, $A$ contains all actions from $\mathcal{A}$ that interfere with $a$ in any state $s' \in \mathcal{S}_{Opt}$.*

Wehrle and Helmert (2014) showed that the cost of an optimal solution does not change if for every state in the state space we only preserve the outgoing transitions that correspond to an action from a strong stubborn set. Put differently, in each state visited during the search we can prune all actions that are *not* in the strong stubborn set, while preserving the guarantee to find optimal solutions.

In practice, it is impossible to efficiently determine a minimal strong stubborn set because we do not know *Opt* and $\mathcal{S}_{Opt}$. However, if C2 and C3 hold for an overapproximation of these sets, they must also hold for the required sets.

Since the set $\mathcal{S}_{Opt}$ cannot be efficiently computed, for C3 it is common to use a state-independent overapproximation of interference. Alkhazraji et al. (2012) and Wehrle et al. (2013) use a purely syntactic criterion: actions $a$ and $a'$ *potentially conflict* in any state if there is a variable $v \in vars(eff(a)) \cap vars(eff(a'))$ such that $eff(a)[v] \neq eff(a')[v]$. Action $a$ *potentially disables* $a'$ if there is a variable $v \in vars(eff(a)) \cap vars(pre(a'))$ such that $eff(a)[v] \neq pre(a')[v]$. Two actions $a$ and $a'$ then *potentially interfere* if they potentially conflict, $a$ potentially disables $a'$, or $a'$ potentially disables $a$. With this definition, two actions potentially interfere if *there exists some* state in which they interfere.

Wehrle and Helmert (2014) strengthen this approach with mutex information: if the preconditions of two actions are mutually exclusive, they cannot both be applicable in a reachable state, so they never interfere in these states.

It is also already intractable to determine whether a given action is an element of *Opt*. We can still determine a necessary enabling set for $a$ and *Opt* by collecting all achievers of an atom that is not true in $s$ but which $a$ depends on. While this set is not minimal, it can be computed efficiently and indeed this is the strategy employed by previous algorithms for constructing strong stubborn sets in planning. Similarly, C1 is satisfied by picking an atom from the goal that is not true in $s$ and including all actions that achieve this atom, hence including at least one action from every plan.

## Existing Action-Centric Algorithm

To satisfy the properties of strong stubborn sets, previous algorithms start from an action set that satisfies C1 and successively add actions to satisfy C2 and C3 until a fixed point is reached. We will adopt the same high-level approach but will differ from this action-centric approach on a lower level.

Before we go into details, we first introduce and analyze the action-centric algorithm. Previously published pseudo-code (Alkhazraji et al. 2012; Al-Khazraji 2017) does not have the level of detail we require for our discussion, but an implementation by Wehrle and Helmert (2014) is available as part of Fast Downward (Helmert 2006). We extracted the pseudo-code as Algorithm 1 from Fast Downward 19.12.[1]

The algorithm collects all actions to be included in the strong stubborn set for a non-goal state $s$ in a collection *stubborn*. The actions for which it still needs to ensure C2 and C3 are tracked in a collection *queue*. To avoid clutter in the pseudo-code, we assume that *stubborn*, *queue* and the components of the task are globally accessible.

---

[1]http://www.fast-downward.org/Releases/19.12

**Algorithm 1** Action-centric algorithm

---

1: **function** COMPUTESTUBBORNSET($s$)
2:     *stubborn* = empty collection
3:     *queue*= empty collection

4:     **procedure** MARKASSTUBBORN($a$)
5:         **if** $a \notin$ *stubborn* **then**
6:             *stubborn.add*($a$)
7:             *queue.add*($a$)

8:     **procedure** ENQUEUEINTERFERERS($a$)
9:         **for** $a' \in \mathcal{A}$ potentially interfering with $a$ **do**
10:            MARKASSTUBBORN($a'$)

11:     **procedure** ENQUEUEACHIEVERS(*atom*)
12:         **for** $a \in \mathcal{A}$ with *atom* $\in$ *eff*($a$) **do**
13:            MARKASSTUBBORN($a$)

14:     *atom* = some unsatisfied goal atom
15:     ENQUEUEACHIEVERS(*atom*)
16:     **while** *queue* is not empty **do**
17:         $a$ = *queue.pop*()        ▷ any element
18:         **if** $a$ is applicable in $s$ **then**
19:            ENQUEUEINTERFERERS($a$)
20:         **else**
21:            ▷ Enqueue a necessary enabling set for $a$
22:            *atom* = some unsatisfied atom from *pre*($a$)
23:            ENQUEUEACHIEVERS(*atom*)
24:     **return** *stubborn*

---

The overall process for generating a strong stubborn set starts with collecting a set of actions to satisfy C1 (lines 14–15). As long as the other conditions are not yet guaranteed for some action $a$ (lines 16–17), it includes further actions to ensure C2 (lines 20–23) or C3 (lines 18–19), depending on whether $a$ is applicable in state $s$ or not.

Whenever an action should be included in the result (marked as stubborn), the algorithm checks if it has already been included previously and if not includes it and enqueues it for further processing into *queue* (lines 4–7).

As mentioned above, necessary enabling sets are generated by starting from an atom and collecting all actions that achieve it (lines 11–13).

## Complexity Analysis

In the complexity analysis, we use $p_{max}$ for the maximal size of a partial state occurring as precondition or effect of any action. In typical planning tasks, this is quite a low number. In general, it can be bounded by the number $|\mathcal{V}|$ of variables.

For an efficient implementation of Algorithm 1, we assume that all state-independent information is precomputed and stored once for every task (i.e., only once for the entire search, not once for every node that is expanded). This affects the set of achievers for every atom (used in line 12) and the interference relation (used in line 9).

The achievers can be determined by one pass over all actions that scans the effect and registers the action accord-

ingly. This requires time $O(|\mathcal{A}|p_{max})$ and the result can be stored in space $O(|\mathcal{P}||\mathcal{A}|)$.

Exploiting pre-sorted action preconditions and effects, the interference relation can be computed in $O(|\mathcal{A}|^2 p_{max})$, ranging over all pairs of actions and syntactically testing their potential interference in time $O(p_{max})$. With no influence on Big-O, we can halve the effort by exploiting that the relation is symmetric. The result can be stored in $O(|\mathcal{A}|^2)$. Fast Downward uses a lazy implementation that only performs the computation for an action once it is required.

We now analyze the time complexity of a single call of COMPUTESTUBBORNSET. With suitable data structures for *stubborn* (e.g., a bitset) and *queue* (e.g., an array-based stack), MARKASSTUBBORN takes constant time. Then ENQUEUEINTERFERERS takes time $O(|\mathcal{A}|)$ because there are at most $|\mathcal{A}|$ interfering actions which have been precomputed. Analogously, ENQUEUEACHIEVERS runs in $O(|\mathcal{A}|)$.

In lines 14 and 22 the algorithm selects an unsatisfied atom from a partial state. Wehrle and Helmert (2014) discussed several such *atom selection strategies*—taken from the literature and new—with different time requirements. To stay general, we account for them with $O(t)$, resulting in time $O(t + |\mathcal{A}|)$ for lines 14–15.

Each iteration of the while loop takes time $O(p_{max})$ for testing applicability plus $O(t+|\mathcal{A}|)$ accounting for the more expensive else-case of the if statement. As every action is added to *queue* at most once, the overall runtime of COMPUTESTUBBORNSET is $O(|\mathcal{A}|(p_{max} + t + |\mathcal{A}|))$. The space complexity for *stubborn* and *queue* is $O(|\mathcal{A}|)$.

## New Atom-Centric Algorithm

The original fixed-point computation from Algorithm 1 tracks (in *queue*) actions that have already been included in the stubborn set but for which it is not yet sure that C2 and C3 are satisfied.

We now reconsider the overapproximation of the interference relation and necessary enabling sets and what this implies for the computation of strong stubborn sets. We begin with potential interference. Using the notion of sibling atoms, we can paraphrase the set of actions that potentially interfere with action $a$: it consists of all actions $a'$ s.t.

- $a'$ achieves a sibling of an atom in *pre*($a$)
  ($a'$ potentially disables $a$), or

- $a'$ depends on a sibling of an atom in *eff*($a$)
  ($a$ potentially disables $a'$), or

- $a'$ achieves a sibling of an atom in *eff*($a$)
  ($a$ and $a'$ potentially conflict).

**Observation 1:** We can characterize these actions by only considering the occurrence of individual atoms in their precondition or effect.

**Observation 2:** The same is true for the actions in the necessary enabling set.

**Observation 3**: The order in which the actions are processed is not important for the fixed-point computation.[2]

---

[2]The order can influence *dynamic* atom selection strategies, but we are not aware of any work that aims for a specific order.

**Algorithm 2** Atom-centric algorithm

---

1: **function** COMPUTESTUBBORNSET($s$)
2:     $stubborn$ = empty collection
3:     $todo\_achievers$, $todo\_dependers = \emptyset$
4:     $seen\_for\_achievers$, $seen\_for\_dependers = \emptyset$

5:     **procedure** HANDLEACTION($a$, $s$)
6:       **if** $a \notin stubborn$ **then**
7:         $stubborn.add(a)$
8:         **if** $a$ is applicable in $s$ **then**
9:           ENQUEUEINTERFERERS($a$)
10:         **else**
11:           ▷ Enqueue a necessary enabling set for $a$
12:           $atom$ = an unsatisfied atom from $pre(a)$
13:           ENQUEUEACHIEVERS($atom$)

14:     **procedure** ENQUEUEACHIEVERS($atom$)
15:       **if** $atom \notin seen\_for\_achievers$ **then**
16:         $todo\_achievers.add(atom)$
17:         $seen\_for\_achievers.add(atom)$

18:     **procedure** ENQUEUEDEPENDERS($atom$)
19:       **if** $atom \notin seen\_for\_dependers$ **then**
20:         $todo\_dependers.add(atom)$
21:         $seen\_for\_dependers.add(atom)$

22:     **procedure** ENQUEUEINTERFERERS($a$)
23:       **for** $atom \in pre(a)$ **do**
24:         **for** all siblings $atom'$ of $atom$ **do**
25:           ENQUEUEACHIEVERS($atom'$)
26:       **for** $atom \in \textit{eff}(a)$ **do**
27:         **for** all siblings $atom'$ of $atom$ **do**
28:           ENQUEUEACHIEVERS($atom'$)
29:           ENQUEUEDEPENDERS($atom'$)

30:     $atom$ = some unsatisfied goal atom
31:     ENQUEUEACHIEVERS($atom$)
32:     **while** $todo\_achievers$ is not empty or
33:        $todo\_dependers$ is not empty **do**
34:       **if** $todo\_achievers$ is not empty **then**
35:         $atom = todo\_achievers.pop()$
36:         **for** $a \in \mathcal{A}$ with $atom \in \textit{eff}(a)$ **do**
37:           HANDLEACTION($a$, $s$)
38:       **else**
39:         $atom = todo\_dependers.pop()$
40:         **for** $a \in \mathcal{A}$ with $atom \in pre(a)$ **do**
41:           HANDLEACTION($a$, $s$)
42:     **return** $stubborn$

---

Based on these three observations, we propose the atom-centric Algorithm 2. The core idea is to achieve synergy effects by deferring the inclusion of actions in the stubborn set, instead tracking the atoms that characterize them.

We use two collections for this purpose: $todo\_achievers$ contains atoms for which all achievers should get included in the stubborn set, $todo\_dependers$ contains atoms for which all actions that depend on the atom should get included.

We ensure that every atom gets added to each of these collections at most once by tracking in sets $seen\_for\_achievers$ and $seen\_for\_dependers$ what has already been included earlier. ENQUEUEACHIEVERS demonstrates this for $todo\_achievers$.

ENQUEUEINTERFERERS and ENQUEUEACHIEVERS play exactly the same role as in the action-centric algorithm, with the only difference that they do not directly mark actions stubborn but instead mark the corresponding atoms for further processing. This directly translates to the initialization of the algorithm in lines 30 and 31.

The main loop (lines 33–41) processes the atoms from $todo\_achievers$ and $todo\_dependers$, initiating the previously deferred handling of actions. HANDLEACTION adds the action to the stubborn set and triggers the later inclusion of interfering actions and necessary enabling sets to satisfy C2 and C3.

**Theorem 1.** *Function* COMPUTESTUBBORNSET($s$) *returns a strong stubborn set for $s$.*

*Proof sketch.* Including all achievers of an unsatisfied goal atom ensures C1. Whenever set *stubborn* is extended in HANDLEACTION, this procedure initiates the inclusion of actions to satisfy C2 and C3. The overall fixed-point computation guarantees that these are indeed included before termination. □

**Complexity Analysis**

For the analysis, we use $p_{\max}$ as before and $d_{\max}$ for the maximal size of all variable domains.

An efficient implementation of the algorithm precomputes the achievers and dependers of each atom (used in lines 36 and 40) once for the entire search. As discussed in the analysis of the action-centric algorithm, this requires time $O(|\mathcal{A}|p_{\max})$ and space $O(|\mathcal{P}||\mathcal{A}|)$ for storing the result. In contrast to the action-centric algorithm, we do not need to compute and store the interference relation.

With suitable data structures, ENQUEUEACHIEVERS and ENQUEUEDEPENDERS take constant amortized time. As each outer loop of ENQUEUEINTERFERERS iterates over at most $p_{\max}$ atoms and the inner loop over all but one atom for each of these variables, the procedure runs in $O(p_{\max}d_{\max})$. Again using $t$ for the variable selection time, it is then easy to see that HANDLEACTION runs in time $O(p_{\max}d_{\max} + t)$ for actions that are not yet contained in *stubborn* and $O(1)$ for actions already contained in *stubborn*.

For the fixed-point iteration, each atom can be inserted into $todo\_achievers$ at most once and into $todo\_dependers$ at most once, causing runtime $O(|\mathcal{P}|)$ for all parts of the fixed-point loop except the inner loops (lines 36–37 and 40–41).

The runtime for the inner loops can be bounded by the total time spent inside HANDLEACTION. Every action can

| time | Action-centric | Atom-centric |
|---|---|---|
| Precomp. | $O(|\mathcal{A}|^2 p_{max})$ | $O(|\mathcal{A}|p_{max})$ |
| Per node | $O(|\mathcal{A}|^2 + |\mathcal{A}|(p_{max} + t))$ | $O(|\mathcal{A}|(p_{max}d_{max} + t) + |\mathcal{P}|)$ |

| space | Action-centric | Atom-centric |
|---|---|---|
| Precomp. | $O(|\mathcal{A}|^2 + |\mathcal{P}||\mathcal{A}|)$ | $O(|\mathcal{P}||\mathcal{A}|)$ |
| Per node | $O(|\mathcal{A}|)$ | $O(|\mathcal{A}| + |\mathcal{P}|)$ |

Table 1: Overview of complexity results.

only be added to *stubborn* once, giving an upper bound of $O(|\mathcal{A}|(p_{max}d_{max} + t))$ for the calls to HANDLEACTION that add actions to the stubborn set. Each other call takes constant time, and we can bound the total number of such calls by $O(|\mathcal{A}||p_{max}|)$: across the execution of the algorithm, every action is considered in lines 36–37 at most once for each of its effects and in lines 40–41 at most once for each of its preconditions. Hence, the overall runtime of COMPUTES-TUBBORN is $O(|\mathcal{A}|(p_{max}d_{max} + t)) + |\mathcal{P}|$.

The space complexity for *todo_achievers*, *todo_dependers*, *seen_for_achievers* and *seen_for_dependers* is $O(|\mathcal{P}|)$, for *stubborn* it is $O(|\mathcal{A}|)$.

Table 1 shows an overview of all complexity results. The new algorithm clearly dominates the old one in the time and space requirements for the precomputation. For the actual computation of stubborn sets, the new algorithm needs more space, but only linearly in the number of atoms. In the time requirements the algorithms exhibit a very different profile, which lets us expect that the new atom-centric algorithm works better if variable domains are not too large and the task has many more actions than atoms.

## Enhancements

In this section, we discuss two possible enhancements of Algorithm 2. The first one is based on the observation that the algorithm frequently enqueues all siblings of an atom, the second one is a new atom selection strategy.

### Shortcut Handling of all Siblings

Since we frequently add all siblings of an atom to one of the queues, we can expect a number of duplicates. Avoiding this overhead should be particularly beneficial if variable domains are large.

From the perspective of a variable, we can track some compact (incomplete) information on what has already been enqueued, for example in *todo_achievers*. For this purpose, we use a datastructure *marked_achieved* that stores for each variable $v$ one of the following values:

$d \in \mathcal{D}(v)$ representing that all siblings of $(v, d)$ have been enqueued,

$\top$ representing that all atoms for this variable have been enqueued, or

$\bot$ representing that we do not have any such information.

The information is incomplete in the sense that we do not track the inclusion of individual atoms, so the value can for example be $\bot$ or some $d \in \mathcal{D}(v)$ although we have already

seen all atoms for the variable. To update and exploit the stored information, we do not simply enqueue all siblings of *atom* in lines 24–25 and 27–28 of Algorithm 2 but proceed instead as follows:

1: $(v, d) = atom$
2: **if** *marked_achieved*$[v] = \bot$ **then**
3:     **for** all siblings $a$ of *atom* **do**
4:         ENQUEUEACHIEVERS($a$)
5:     *marked_achieved*$[v] = d$
6: **else if** *marked_achieved*$[v] = d' \notin \{d, \top\}$ **then**
7:     ENQUEUEACHIEVERS($(v, d')$)
8:     *marked_achieved*$[v] = \top$

If we do not have sufficient information, we add all siblings of *atom* as before, but remember that all values apart from the one from *atom* have been added (lines 2–5). If we know that all atoms (value $\top$) or all siblings of *atom* (value $d$) have already been added, we do not have to do anything. Otherwise, we add the only missing sibling (whose value is stored in *marked_achieved*$[v]$) and remember that we now have added all atoms (lines 6–8).

We proceed analogously when enqueueing all siblings of an atom with ENQUEUEDEPENDERS in lines 27 and 29.

### Atom Selection Strategy

If an action from the stubborn set is inapplicable, we need to choose an unsatisfied atom from the action precondition as seed for the inclusion of a necessary enabling set. Wehrle and Helmert (2014) already discussed and evaluated several strategies for this choice point.

We want to propose a new strategy, called *quick skip*. It is easy to see that if the chosen atom has already been seen (included in *seen_for_achievers*), the algorithm does not enqueue anything within ENQUEUEACHIEVERS. This saves computational effort and—maybe even more importantly— it can potentially lead to more pruning because we do not unnecessarily grow the stubborn set. Therefore, in line 12 of Algorithm 2 the quick skip strategy chooses some atom from $pre(a) \cap$ *seen_for_achievers* whenever this set is not empty.

This selection strategy is related to the *static small* and *dynamic small* strategies by Wehrle and Helmert, both of which aim to keep the resulting stubborn set small. The static strategy prefers variables that appear in the effects of fewer actions, the dynamic one prefers atoms with a minimal number of achieving actions that have not yet been included in the stubborn set. Our proposed strategy is closer to *dynamic small* but less specific. If there is an atom for which *all* achieving actions have already been scheduled for inclusion, the strategies are equal. Otherwise, our strategy can be combined with any other strategy, leaving another choice point.

## Experimental Evaluation

We implemented the atom-centric algorithm on top of Fast Downward 19.12, which already contains an implementation of the action-centric algorithm (called "simple stubborn sets" there). For the evaluation, we use the benchmarks of all optimal tracks of all International Planning Competitions from 1998 to 2018, amounting to 1827 tasks from 65 domains. Experiments were run on Intel Xeon Silver

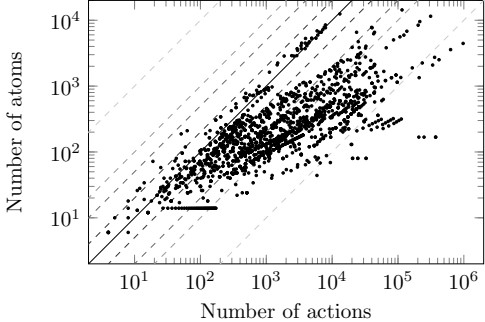

Figure 1: Number of atoms vs. actions for all tasks in the benchmark set. Each mark represents one task. Dashed diagonals show factors 2, 5, 10, and 100.

4114 CPUs using Downward Lab (Seipp et al. 2017). Each planner run is limited to 1800 seconds and 3.5 GiB. The benchmarks, code and experimental data are published online (Röger et al. 2020b).

Before we compare different algorithms and configurations, we evaluate whether the different time complexity of the atom-centric algorithm is promising at all. For this purpose, in Figure 1, we plot the number of atoms against the number of actions in the $SAS^+$ planning tasks produced by Fast Downward. We see that the actions frequently outnumber the atoms, often by several orders of magnitude, so the trade-off looks promising indeed.

## Action- vs. Atom-Centric Algorithm

In the first experiment, we examine how the plain action-centric and atom-centric algorithms compare when they compute the same information. Towards this end, we do not use the mutex-based strengthening of interference and use the same strategy for choosing unsatisfied atoms for both algorithms, namely always picking the first unsatisfied atom according to the fixed variable ordering of Fast Downward.

**Blind Search** With blind search, node expansions are extremely fast, so the relative overhead of computing stubborn sets for each expansion is high. For this reason, we can only expect to benefit from partial order reduction if it leads to significant pruning. On our benchmark set, blind search without pruning solves 710 instances, whereas coverage increases by 26 instances with the atom-centric algorithm (cf. left part of Table 2). Interestingly, computing the same information with the action-centric algorithm leads to a significant coverage decline to 680 instances. As expected, in both cases the total number of expansions is the same and decreases by 17.9% compared to no pruning across all tasks solved by all three configurations.

A closer look at the results per domain reveals that even with our more efficient algorithm, using strong stubborn set pruning is not always beneficial, losing 1–3 tasks in 11 domains and even 5 instances in the freecell domain. The positive net benefit stems from the two parcprinter domains with a coverage increase of 20 and 14 and the two woodworking domains with an increase of 8 and 7 tasks. So it seems that

| | blind | | | LM-cut | | | SCP | | |
|---|---|---|---|---|---|---|---|---|---|
| | base | action | atom | base | action | atom | base | action | atom |
| airport (50) | **22** | 21 | 21 | **28** | **28** | **28** | **24** | **24** | **24** |
| data-network (20) | 7 | 6 | **7** | **12** | **12** | **12** | **14** | 13 | **14** |
| freecell (80) | **20** | 9 | 15 | **15** | **15** | **15** | **68** | 49 | 61 |
| hiking (20) | **11** | 8 | **11** | **9** | **9** | **9** | **14** | 11 | 13 |
| miconic (150) | **55** | 50 | **55** | **141** | **141** | **141** | 143 | **144** | **144** |
| mprime (35) | **19** | 18 | **19** | **22** | **22** | **22** | **31** | 30 | **31** |
| nomystery (20) | 8 | 7 | **8** | **15** | 14 | **15** | **20** | **20** | **20** |
| openstacks-08 (30) | **22** | 20 | **22** | **22** | 20 | **22** | **22** | 20 | **22** |
| openstacks-11 (20) | **17** | 15 | **17** | **17** | 15 | **17** | **17** | 15 | **17** |
| org.-synth.-split (20) | **10** | 9 | 9 | **16** | 15 | 15 | **10** | 9 | 9 |
| parcprinter-08 (30) | 10 | **30** | **30** | 19 | **30** | **30** | 19 | **30** | **30** |
| parcprinter-11 (20) | 6 | **20** | **20** | 14 | **20** | **20** | 15 | **20** | **20** |
| parking-11 (20) | **0** | **0** | **0** | **3** | **3** | **3** | **7** | 4 | **7** |
| parking-14 (20) | **0** | **0** | **0** | **4** | 3 | **4** | **6** | 4 | **6** |
| pegsol-08 (30) | **27** | **27** | **27** | **29** | 28 | 28 | **30** | **30** | **30** |
| pegsol-11 (20) | **17** | **17** | **17** | **19** | 18 | 18 | **20** | **20** | **20** |
| petri-net-align. (20) | **4** | 2 | **4** | **9** | **9** | **9** | **0** | **0** | **0** |
| pipesworld-not. (50) | **17** | 14 | 16 | **18** | **18** | **18** | **24** | **24** | **24** |
| pipesworld-t. (50) | 12 | 8 | 11 | **12** | **12** | **12** | **17** | 12 | 16 |
| rovers (40) | 6 | **7** | **7** | 9 | **10** | **10** | 8 | **9** | **9** |
| satellite (36) | **6** | **6** | **6** | 7 | **12** | **12** | 7 | 8 | **9** |
| scanalyzer-08 (30) | **12** | 8 | 9 | **16** | 14 | 15 | **18** | 16 | **18** |
| scanalyzer-11 (20) | **9** | 5 | 6 | **13** | 11 | 12 | **15** | 13 | **15** |
| snake (20) | **11** | 4 | 9 | **7** | 6 | **7** | **13** | 7 | 11 |
| spider (20) | **11** | 6 | 9 | **11** | **11** | **11** | **16** | 13 | 15 |
| termes (20) | **9** | 6 | **9** | **6** | 5 | **6** | **13** | 11 | **13** |
| tetris (17) | **9** | 6 | 7 | **6** | **6** | 5 | **11** | 9 | 10 |
| tidybot-11 (20) | **13** | 5 | 12 | **14** | **14** | **14** | **15** | 13 | 14 |
| tidybot-14 (20) | **6** | 0 | 4 | **9** | 8 | 8 | **10** | 5 | 9 |
| transport-11 (20) | **6** | **6** | **6** | **6** | **6** | **6** | **13** | 12 | **13** |
| transport-14 (20) | **7** | 6 | **7** | **6** | **6** | **6** | **9** | 8 | 8 |
| trucks (30) | **6** | 5 | **6** | **10** | **10** | **10** | **12** | **12** | **12** |
| woodworking-08 (30) | 8 | **16** | **16** | 18 | **27** | **27** | 26 | **30** | **30** |
| woodworking-11 (20) | 3 | **10** | **10** | 12 | **19** | **19** | 19 | **20** | **20** |
| zenotravel (20) | **8** | 7 | **8** | **13** | **13** | **13** | **13** | **13** | **13** |
| sum (1088) | 414 | 384 | **440** | 587 | 610 | **619** | 719 | 678 | **727** |
| other domains (739) | **296** | **296** | **296** | **373** | **373** | **373** | **417** | **417** | **417** |
| total (1827) | 710 | 680 | **736** | 960 | 983 | **992** | 1136 | 1095 | **1144** |

Table 2: Coverage of A* with the blind (left), LM-cut (middle), and SCP (right) heuristics, comparing vanilla search (base) with the addition of plain action-centric (action) and atom-centric (atom) pruning. We highlight maximum coverage separately for each heuristic.

these domains are especially suitable for partial order reduction, whereas in other domains the additional overhead does not pay off. Indeed, in woodworking the goal is to process a set of work pieces, each basically corresponding to an independent subtask. In parcprinter, the aim is to print a set of pages using several components of an involved printing system. The actions for the different pages can often be arbitrarily interleaved, which can be avoided with partial order reduction.

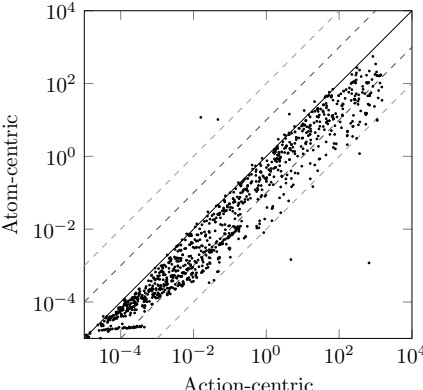

Figure 2: Comparison of pruning time (within an A* search with the SCP heuristic) of the action-centric and the atom-centric algorithm for tasks solved by both approaches. Numbers are in seconds and specify the total time spent computing stubborn sets over all node expansions.

**LM-Cut**  Wehrle and Helmert (2014) used A* search with the LM-Cut heuristic (Helmert and Domshlak 2009) for their evaluation. In this setting, stubborn set pruning is useful overall (cf. middle part of Table 2). 960 tasks are solved without pruning, 983 with the action-centric algorithm and 992 with the atom-centric computation. In a per-domain comparison to the baseline without pruning we never lose more than one task, but coverage increases in six domains. However, this is again most prominent in the parcprinter (+11 and +6) and the woodworking (+9 and +7) domains. The advantage in comparison to the action-centric algorithm stems from eight domains. Conversely, the action-centric variant only solves one more instance in tetris.

**Saturated Cost Partitioning**  We also conducted an analogous experiment for A* with a saturated cost partitioning (SCP) heuristic (Seipp, Keller, and Helmert 2020) over pattern databases (Edelkamp 2001) and Cartesian abstractions (Seipp and Helmert 2018). The pattern databases were generated systematically up to pattern size 2 and via hill climbing (Haslum et al. 2007). This SCP heuristic yields state-of-the-art performance for optimal classical planning, because it is both accurate and very fast to evaluate (much faster than LM-cut, for example).

Similarly to the results for blind search, using the action-centric algorithm decreases coverage (cf. right part of Table 2). In contrast to the results for LM-cut, using the atom-centric algorithm increases the total coverage by only 2 tasks, with a decrease by 1 task in 8 domains, by 2 in snake and even by 7 in freecell. However, if we compare the action-centric against the atom-centric algorithm, we see a clear advantage of the new one in 19 domains, while the opposite is never the case.

**Overall**  As both stubborn set algorithms compute the same information, the difference in performance must be attributed to the different computational overhead. Figure 2 compares the total time spent for computing stubborn sets

| | ds | +sib | ss | +sib | FD | +sib | qs | +sib |
|---|---|---|---|---|---|---|---|---|
| airport (50) | **25** | **25** | 24 | 24 | 24 | 24 | 24 | 24 |
| data-network (20) | 13 | 13 | **14** | **14** | **14** | **14** | **14** | **14** |
| freecell (80) | 42 | 43 | 60 | 60 | **61** | **61** | 59 | 60 |
| ged (20) | 15 | 15 | **19** | **19** | **19** | **19** | **19** | **19** |
| hiking (20) | 11 | 11 | 12 | 12 | **13** | **13** | 12 | 12 |
| logistics98 (35) | 12 | 12 | 12 | 12 | 12 | 12 | **13** | **13** |
| miconic (150) | **144** | 143 | **144** | **144** | **144** | **144** | 143 | **144** |
| mprime (35) | 30 | 30 | 30 | 30 | **31** | 30 | 30 | 30 |
| mystery (30) | 18 | 18 | **19** | **19** | **19** | **19** | **19** | **19** |
| openstacks-08 (30) | 20 | 20 | 22 | 22 | 22 | 22 | **23** | **23** |
| openstacks-11 (20) | 15 | 15 | 17 | 17 | 17 | 17 | **18** | **18** |
| openstacks-14 (20) | 3 | 3 | 3 | 3 | 3 | 3 | **4** | **4** |
| parcprinter-08 (30) | **30** | **30** | 28 | 28 | **30** | **30** | **30** | **30** |
| parcprinter-11 (20) | **20** | **20** | 18 | 18 | **20** | **20** | **20** | **20** |
| parking-11 (20) | 4 | 4 | **7** | **7** | **7** | **7** | **7** | **7** |
| parking-14 (20) | 4 | 4 | **6** | **6** | **6** | **6** | **6** | **6** |
| pipesworld-not. (50) | 21 | 21 | **24** | **24** | **24** | **24** | **24** | **24** |
| pipesworld-t. (50) | 11 | 11 | 13 | 13 | **16** | **16** | 14 | 16 |
| rovers (40) | **11** | **11** | 10 | 10 | 9 | 9 | 9 | 9 |
| scanalyzer-08 (30) | 15 | 15 | **18** | **18** | **18** | **18** | **18** | **18** |
| scanalyzer-11 (20) | 12 | 12 | **15** | **15** | **15** | **15** | **15** | **15** |
| snake (20) | 4 | 4 | 10 | 10 | **11** | **11** | 10 | **11** |
| spider (20) | 12 | 12 | 14 | 14 | **15** | **15** | **15** | **15** |
| termes (20) | 10 | 10 | 12 | 12 | **13** | **13** | **13** | **13** |
| tetris (17) | 8 | 8 | **10** | **10** | **10** | **10** | **10** | **10** |
| tidybot-11 (20) | 12 | 13 | **14** | **14** | **14** | **14** | **14** | **14** |
| tidybot-14 (20) | 4 | 4 | 8 | 8 | **9** | **9** | **9** | **9** |
| transport (20) | 10 | 10 | **13** | **13** | **13** | **13** | **13** | **13** |
| sum (927) | 536 | 537 | 596 | 596 | 609 | 608 | 605 | **610** |
| other domains (900) | **535** | **535** | **535** | **535** | **535** | **535** | **535** | **535** |
| total (1827) | 1071 | 1072 | 1131 | 1131 | 1144 | 1143 | 1140 | **1145** |

Table 3: Coverage of A* with the SCP heuristic, comparing atom selection strategies *ds*, *ss*, *FD*, and *qs*, and the combination with shortcut handling of siblings (*sib*).

for each task. This data stems from the experiment with the SCP heuristic; the plots for the other configurations (blind and LM-Cut) look similar. We see that with the atom-centric algorithm we can obtain the same pruning power much faster, often by more than an order of magnitude.

### Enhancements to the Action-Centric Algorithm

In the second experiment, we evaluate the two enhancements to the atom-centric algorithm. The shortcut handling of siblings (*sib*) does not change the behavior of the algorithm and should hence only have an impact on runtime. The new atom selection strategy quick skip (*qs*) should have a tendency to produce smaller stubborn sets, so we would also hope for more pruning. We compare the *qs* strategy to the default strategy (*FD*) and the two strategies dynamic small (*ds*) and static small (*ss*) by Wehrle and Helmert (2014), all of them with and without the *sib* enhancement.

Table 3 shows that with the SCP heuristic, the dynamic small and static small strategies solve many fewer tasks than

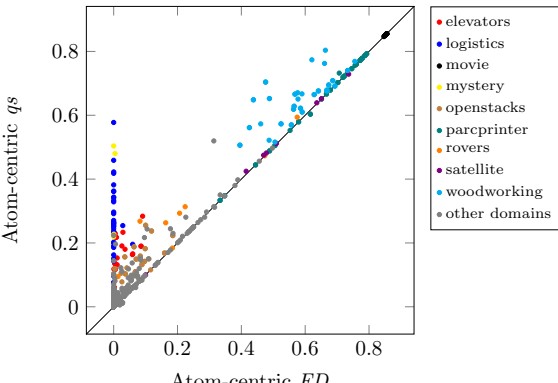

Figure 3: Comparison of pruning ratio of the atom-centric algorithm with strategies *FD* and *qs*, using A* with SCP.

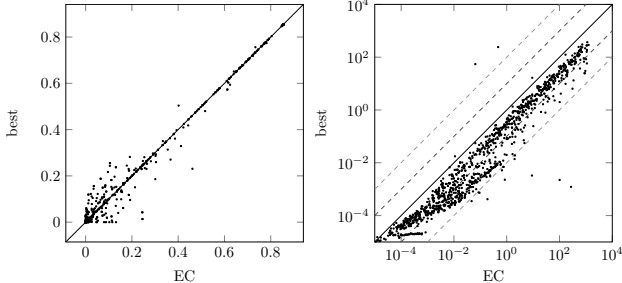

Figure 4: Comparison of pruning ratio (left) and pruning time (right) of EC vs. our best configuration with SCP.

the Fast Downward and quick skip strategies. While the Fast Downward strategy solves 4 more tasks in total than quick skip, quick skip is better than Fast Downward with the LM-cut and blind heuristics (+3 and +2, not shown in Table 3). The shortcut handling of siblings only has a very mild impact on coverage, sometimes negative, sometimes positive, except for the new strategy quick skip where it is exclusively beneficial (except for transport-14 with the blind heuristic, not shown in Table 3). The combination of quick skip with shortcut handling of siblings achieves the highest total coverage with all three heuristics, and dominates the other strategies also in a per-domain comparison, except for 7 domains when using blind search, 3 domains when using A* with LM-cut, and 5 domains when using A* with SCP.

To analyze the pruning ratio of the different methods, we run the search with pruning and accumulate the number of successors of all expanded states as $n_{all}$ and sum up the size of the corresponding stubborn sets as $n_{gen}$. The pruning ratio is then defined as $1 - n_{gen}/n_{all}$, giving values between 0 and 1, where 0 represents no pruning and 1 would mean that all successors were pruned. Figure 3 plots the pruning ratio of the atom-centric algorithm with the *FD* strategy (the best previous selection strategy according to Table 3) to the new quick skip strategy (both with SCP), highlighting domains with larger differences. We observe consistent positive impact on the pruning power, which is particularly pronounced in the logistics and woodworking domains.

## Comparison to the State of the Art

In the third experiment, we compare our best configuration (atom-centric algorithm with *qs* and *sib*) to the configuration reported as state of the art for computing strong stubborn sets by Wehrle and Helmert (2014), namely "SSS-EC full/mutex" (EC), which computes stubborn sets in a way that dominates the expansion core method (Chen and Yao 2009) and enhances action interference with mutexes. With our best configuration, total coverage increases significantly for all three heuristics, in particular for the two faster-to-compute ones (+51 with blind search, +9 with LM-Cut, +38 with SCP). A deeper analysis reveals that our configuration

is on-par with respect to pruning power, but requires a much lower computation time. To illustrate this, we compare the pruning ratio and pruning time for A* with SCP in Figure 4. For the other heuristics the results look qualitatively similar.

Since not all domains are equally suited for partial-order reduction, many recent IPC planners (e.g., Alkhazraji et al. 2014) disable pruning if after 1000 expanded states the pruning ratio is at most 20%. We evaluated the impact of this approach on our best configuration (atom-centric algorithm with both enhancements) and on the previous state of the art (EC). The method has only a mild impact on our algorithm: overall, coverage increases, but in some domains fewer tasks are solved. This is very different for the slower EC method, which greatly benefits from this approach, bringing it almost on par with our configuration.

## Discussion and Future Work

We proposed an atom-centric algorithm that computes the same stubborn sets as an earlier action-centric algorithm with a different profile time and space complexity profile. The new algorithm requires less space, and we saw that it is much faster on common planning benchmarks. One limitation of our algorithm is that it is no longer possible to enhance the interference relation with mutex information. However, already without any enhancements, our algorithm outperforms the best previous algorithm (EC), which makes use of such mutex information (1145 vs. 1107 solved tasks with A* + SCP). The new atom selection strategy *quick skip* does not only further speed up the computation but also leads to smaller stubborn sets and thus to more pruning.

In classical planning, stubborn sets have not only been used for state-space search but have also been adapted for fork-decoupled search (Gnad, Hoffmann, and Wehrle 2019). Beyond the classical planning fragment, they have been applied to fully observable non-deterministic planning (Winterer et al. 2017), planning with resources (Wilhelm, Steinmetz, and Hoffmann 2018) as well as for goal recognition design (Keren, Gal, and Karpas 2018). In future work, it would be interesting to examine whether the general idea of an atom-centric perspective can also be beneficially applied in these settings.

## Acknowledgments

We have received funding for this work from the European Research Council (ERC) under the European Union's Horizon 2020 research and innovation programme (grant agreement no. 817639).

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
