# OpenReview forum: "An Atom-Centric Perspective on Stubborn Sets"
_icaps-conference.org/ICAPS/2020/Workshop/HSDIP — HSDIP 2020_

### Official Review · AnonReviewer1 · 2020-03-31
**Initial assessment**

**Rating:** 8
**Confidence:** 4

**Review:**

The paper presents an interesting improvement of the implementation of strong
stubborn sets. Since this supposed to be only an initial assessment, I didn't go over
every detail of the paper, but I think it is a solid paper. I noticed one typo
in the "Complexity Analysis" secion: O(|ops|). And what looks suspicious is
3.5 GB memory. Why 3.5 and not 4 or, better, 8 GB?

---

> ### Author Response · Authors · 2020-04-03
> **Why 3.5 GiB**
>
> We have 4 GiB of physical memory available for each planner run and need to leave some of the memory to the operating system. We have seen this amount of memory used many times in the planning literature (for the same reason). Thanks for pointing out the typo!

---

> ### Comment · AnonReviewer1 · 2020-08-16
> **Review: Clear accept**
>
> I read the (updated) paper again and I don't see anything that should be
> addressed by the authors. The paper describes a more efficient algorithm for
> computing strong stubborn sets than is known from literature (and from the
> implementation in FD). The paper is written clearly with a lot of details that
> are easy to follow, the complexity analysis seems to be correct, and the
> experimental evaluation is comprehensive. So, I increased my rating to 8.
>
> I have only one note/question (possibly for a future work): The action-centric
> algorithm can be enhanced with mutexes (as you clearly state in the text and
> then compare to it in the experimental section), but utilizing mutexes in your
> approach doesn't seem to be possible (or at least not directly without some
> additional machinery). Do you think constructing FDR variables differently
> could significantly change the outcome? Intuitively, it seems to me that the
> overapproximation approach with "potential interference" may be sensitive to the
> construction of FDR variables, but it also seems to me that using mutexes
> could compensate for selecting "bad" variables. It would be interesting to see
> whether we can construct FDR variables in a such a way that would be helpful
> to the inference of SSS.

---

### Author Response · Authors · 2020-06-23
**Updated version**

We updated the PDF. The main difference is a tighter complexity analysis of the new algorithm but we also improved the presentation throughout the paper.

---

### Comment · Program_Chairs · 2020-09-14
**Final Decision: Accept**

Dear Authors,

Thank you very much for your submission. We are happy to inform you that we have decided to accept it and we look forward to your talk in the workshop. You will receive additional information per mail in the coming days.

Best,
The HSDIP'20 team

---

### Decision · Program_Chairs · 2020-09-30

Accept